# Examining the role of systemic inflammation as a mediator of the glycaemia-brain volume associations in women

Nasri Fatih [1,2]*, Nish Chaturvedi[2], Victoria Garfield[3], Carole H. Sudre[2],
Richard J. Silverwood[4], David M. Cash[5,6], Ian B. Malone[5], Josephine Barnes[5],
Marcus Richards[2], Jonathan M. Schott[5,6ᵒ], Alun D. Hughes[2ᵒ], Sarah-Naomi James[2,5ᵒ]

1 Nuffield Department of Population Health, Big Data Institute, University of Oxford, Oxford, United
Kingdom, 2 Unit for Lifelong Health and Ageing at UCL, London, United Kingdom, 3 Department of
Pharmacology and Therapeutics, University of Liverpool, Liverpool, United Kingdom, 4 Centre for
Longitudinal Studies, UCL Social Research Institute, University College London, London, United Kingdom,
5 Dementia Research Centre, UCL Queen Square Institute of Neurology, University College London,
London, United Kingdom, 6 UK Dementia Research Institute at UCL, University College London, London,
United Kingdom

ᵒ These authors contributed equally to this work.
* nasri.fatih@ndph.ox.ac.uk

journal.pone.0329046

Hawai'i at Manoa College of Tropical Agriculture
and Human Resources, UNITED STATES OF
AMERICA

**Peer Review History:** PLOS recognizes the
benefits of transparency in the peer review
process; therefore, we enable the publication
of all of the content of peer review and
author responses alongside final, published
articles. The editorial history of this article is
available here: https://doi.org/10.1371/journal.
pone.0329046

## Abstract

Previous studies have found that diabetes and its mechanistic factors (e.g., glycae-
mia) are associated with poorer cognitive and brain health. There is also growing
evidence of sex differences in how diabetes manifests itself and impacts the brain.
The mechanisms through which this association manifests itself are still poorly under-
stood, but the possible role of inflammation has been proposed. This study aims to
explore whether the relationship between mid-life glycaemia and brain volumes in
later-life in women is mediated by systemic inflammation. The sample consisted of
female participants from the National Survey of Health and Development (NSHD)
who underwent neuroimaging as part of the Insight 46 sub-study. Path analysis
models were then constructed between glycaemic markers (age 60−64) and brain
health outcomes (age 69−71) with adjustments for social and metabolic confound-
ers (age 60−64). Although fasting glucose was associated with higher GlycA levels
(β = 0.05 [0.01, 0.10], p = 0.005), associations with CRP and IL-6 were weaker and
not statistically significant (e.g., IL-6: β = 0.10 [−0.04, 0.30], p = 0.102). However, we
did not find evidence that inflammatory markers were associated with brain volume
outcomes (e.g., IL-6 and whole brain volume: β = −3.4 [−8.1, 1.3], p = 0.092; IL-6 and
grey matter volume: β = −0.4 [−1.9, 1.0], p = 0.512). Consequently, indirect (mediated)
effects via systemic inflammation were not observed. This suggests that alternative
mechanisms beyond inflammation may contribute to the relationship between mid-life
glycaemia and later-life brain health.

**Data availability statement:** Data from the MRC National Survey of Health and Development (NSHD) are curated by the MRC Unit for Lifelong Health and Ageing at UCL and are available to bona fide researchers via application through the NSHD data-sharing platform (Skylark: www.skylark.ucl.ac.uk/NSHD). An NSHD dataset is also deposited with the UK Data Service (DOI: 10.5255/UKDA-SN-8732-3).

**Funding:** This work was supported by Alzheimer's Research UK (awards ARUK-PG2014- 1946 and ARUK-PG2017-1946 to Jonathan Schott; additional support to Josephine Barnes), Alzheimer Society (award AS-JF-17-011 to Carole Sudre), UCL Queen Square Institute of Neurology, University College London (support to Jonathan Schott), Diabetes Research and Wellness Foundation (award SCA/01/NCF/22 to Victoria Garfield), Medical Research Council funding for the Medical Research Council National Survey of Health and Development (MC_UU_12019/1), British Heart Foundation (PG/17/90/33415) to Alun Hughes. Alun Hughes also receives support from the Horizon Europe Programmes of the European Union through Innovate UK (HORIZON-RIA 10113672), the Wellcome Trust (221774/Z/20/Z), a British Heart Foundation Centre of Research Excellence award to UCL (RE/24/130013) and the National Institute for Health Research University College London Hospitals Biomedical Research Centre The funders of the study had no role in study design, data collection, analysis, or interpretation, or report writing. All authors had full access to all the data in the study. The corresponding author had final responsibility for the decision to submit for publication.

**Competing interests:** NC serves on Data Monitoring and Safety Committees for trials sponsored by AstraZeneca. JS has received research funding from Avid Radiopharmaceuticals (a wholly owned subsidiary of Eli Lilly), has consulted for Roche Pharmaceuticals, Biogen, and Eli Lilly, given educational lectures sponsored by GE, Eli Lilly and Biogen, and serves on a Data Safety Monitoring Committee for Axon Neuroscience SE. The remaining authors declare that there are no relationships or activities that might bias, or be perceived to bias, their work.

## Introduction

Previous studies have shown an association between diabetes and poorer cognitive health as well as higher risk of dementia. In our previous work, we found that mid-life glycaemia (as indexed by HbA1c) was associated with poorer brain health later in life exclusively in women. [1,2]. This aligns with other studies which also report sex-specific differences in diabetes-related mechanisms and health outcomes [3].

An important next step that follows from these findings is to investigate potential mechanistic explanations for these sex differences. One potential mediating factor of this relationship could be inflammation since there is growing evidence of: 1) evidence that inflammation is a central feature of type 2 diabetes (T2D) [4] and 2) associations between systemic inflammatory markers such as interleukin-6 (IL-6) and brain health outcomes both in animal models and humans [5–7] 3) higher systemic inflammation in women [8]. Sex differences in inflammation levels may particularly become more prominent and influential following the decline of oestrogen's neuroprotective and anti-inflammatory effect during the perimenopause and following the menopause [9,10]. A sex-specific transcriptomic analysis also revealed that female microglia exhibit evidence of both higher levels of inflammation and Alzheimer's disease-related (AD) genes compared to male microglia suggesting a higher neuroinflammatory response in women, which may have broader implications for sex differences in brain health and disease [11]. Thus, systemic inflammation may mediate the sex-specific associations between glycaemia, and volumetric brain health predominantly observed in women.

Here we expand on our previous findings and investigate the extent through which the relationship between glycaemic markers (HbA$_{1c}$ and glucose) at age 60−64 and volumetric brain measures at age ～70 in female participants was mediated by systemic inflammation. Inflammation was indexed by IL-6, C-reactive protein (CRP) and GlycA, a spectroscopic marker of systemic inflammation related to the levels and degree of glycosylation of various acute phase proteins.

## Materials and methods

### Sample and ethics

The sample consisted of participants of the National Survey of Health and Development (NSHD). NSHD is a British birth cohort originally made up of 5,362 boys and girls born across mainland Britain during the same week in 1946 [12]. In 2006, study members were sent postal questionnaires. They were also invited to attend a clinic visit between ages 60 and 64 (conducted over multiple years), during which blood samples were collected to assess glycaemic health and systemic inflammation.

Between 2015–2018, a subset of 502 NSHD participants were enrolled into the Insight 46 sub-study to undergo neuroimaging and further assessments. Selection was restricted to those who had previously attended the clinic-based assessment at age 60–64, had previously intimated they were willing to attend a clinic visit in London and for whom relevant data in childhood and adulthood are available [13].

The study protocol—including this secondary analysis of NSHD data—was reviewed and approved before any data were accessed by the London-Queen Square NHS Research Ethics Committee (14/LO/1073) and by the Scotland A Research Ethics Committee (14/SS/1009). All NSHD waves obtain written informed consent from participants in accordance with the Declaration of Helsinki. The dataset supplied to the authors for the present analysis was fully pseudonymised; direct identifiers are retained on a separate, access-controlled server. The lead author first accessed the anonymised extract for research purposes on 3 March 2022.

## Investigations

**Exposure variables.** $HbA_{1c}$ and glucose were measured in a fasting blood sample collected at age 60–64. $HbA_{1c}$ was measured by ion exchange High-performance liquid chromatography (HPLC) on a Tosoh analyzer (Tosoh Bioscience, Tessenderlo, Belgium) whereas glucose was measured via an enzymatic assay using hexokinase coupled to glucose 6-phosphate dehydrogenase on a Siemens Dimension Xpand analyzer (Siemens Medical Solutions, Erlangen, Germany). *Systemic inflammation mediators*: Inflammatory markers were measured from fasting blood samples collected at age 60–64 during clinic or home visits. Interleukin 6 (IL-6)was assayed from serum using ELISA (inter-assay CV: 6.5%) and reported in pg/L. CRP was measured using a high-sensitivity particle-enhanced immunoturbidimetric assay (CV: 6.28%, detection limit: 1 ng/ml) and reported in mg/L. Glycoprotein acetyls (GlycA), specifically alpha1-acid glycoprotein, were analyzed via high-throughput nuclear magnetic resonance (NMR) metabolomics on serum samples (no freeze-thaw cycles) using Bruker spectrometers, with most inter-assay CVs below 5%, as detailed in Soininen et al. (2015).

**Confounders.** Confounders were identified based on prior knowledge of associations between hyperglycaemia and inflammation, and inflammation and brain health, which were then represented through a directed acyclic graph (DAG) (see S1B Fig). Confounders were considered for both the exposure-mediator relationships and the mediator-outcome relationships. In this analysis, the confounders considered were:

**Socioeconomic position:** Childhood socioeconomic position (SEP) was measured as father's occupational social class recorded at age 4 (or if missing, at age 11) and categorised into manual or non-manual according to the UK Registrar General's Standard's Occupation Classification. Adult SEP was based on head of household occupation at age 53 years and categorised into manual or non-manual.

**Education:** The highest educational attainment or training qualification achieved by 26 years was classified according to the Burnham scale [14] and grouped into the following: no qualification; below ordinary secondary qualifications (e.g., vocational qualifications); ordinary level qualifications ('O' levels or their training equivalents); advanced level qualifications ('A' levels or their equivalents); or higher education (degree or equivalent).

**Alcohol:** Information on alcohol consumption over the previous 7 days was obtained by a self-completed questionnaire completed between the ages of 60–64. Responses were totalled to provide an approximate measure of drinks per week, where a drink (or unit in UK terminology) contains ~9.0 g of alcohol. Participants were then dichotomised into two categories: those who drunk under 14 units of alcohol per week and those who drunk 14 or more units of alcohol per week.

**Smoking:** Smoking status at age 60–64 was assessed by self-report and was classified into three groups: current smokers, ex-smokers, and never-smokers.

**Physical activity:** Physical activity information was collected at age 60–64 using the EPIC physical Activity questionnaire-2 [15]. This assessed how often participants had had taken part in any sports, vigorous leisure activities or exercise in the previous 4 weeks. Responses were categorised into: 1) not active (no participation in physical activity/month), 2) moderately active (participated 1–4 times/month) and 3) most active (participated 5 or more times/month) [16].

**Body mass index:** body mass index was calculated using the following equation: (weight(kg)/height(m$^2$)). Height and weight (at age 60–64) were measured by nurses using standardised protocols.

**Arthritis:** Arthritis status was assessed through questionnaires asking whether participants had taken non-steroidal anti-inflammatory medication at age 60–64 or they had been diagnosed with the condition.

**COPD:** Chronic obstructive pulmonary disease (COPD) at age 60–64 was identified based on the presence of airflow obstruction defined by the ratio of $FEV_1$/FVC of less than the lower limit of normal or 0.7 [17].

**Volumetric neuroimaging outcomes.** We ascertained metrics of whole brain volume (WBV), grey matter (GM) and white matter (WM) volumes using validated pipelines. In brief, individuals underwent MRI scanning on the same Biograph mMR 3T PET-MRI scanner (Siemens Healthcare, Erlangen), which included high resolution 3D (1·1 mm isotropic) T1-weighted and T2-weighted Fluid attenuated inversion recovery (FLAIR) sequences as previously described [18,19].

Following closely the pipelines detailed in Eshagi and colleagues [20], original parcellation applying GIF [21]was used to identify WMH (BaMoS). These lesion maps were used to inpaint the T1 weighted [22] image with healthy looking tissue before performing the parcellation on the healthy looking image to avoid any possible processing bias introduced by the lesions. WBV was segmented using Multi-Atlas Propagation and Segmentation (MAPS) [23].

## Statistical analyses

To address the extent to which systemic inflammation mediates the relationship between glycemia and brain volume, we conducted a two-component mediation analysis using the 'sem' package (StataCorp) in Stata 17 (StataCorp, College Station, TX, USA). This approach, which is analogous to path analysis, decomposes the total 'effect' (i.e., association between exposure and outcome) into a direct, and an indirect (or mediated) effect [24,25] (see S2 Fig). While a three-component mediation analysis which also included a mediated interactive effect was contemplated, for reasons that will become apparent from the results, this was not performed. Note that in this analysis framework the term "effect" is used to describe the relationships between exposure, mediator and outcome. Such relationships or associations can only be interpreted as causal under very strong assumptions that are arguably never (or perhaps almost never) satisfied. Some eschew the term 'effect' in all epidemiological studies for this reason, but we emphasise that casual interpretations should not be applied to the term "effect" in this context. This is discussed by Hernan and colleagues [26,27].

All models were estimated using full information maximum likelihood (FIML), which assumes linear relationships and multivariate normality but allows for missing data under a missing at random (MAR) assumption and was therefore considered preferable to estimation based on complete case data. As a sensitivity analysis, we also conducted a complete case analysis using an estimator assuming multivariate normality to assess whether violations of normality due to categorical or missing data handling in FIML affected our results.

Due to their skewed distribution, IL-6 and CRP were log-transformed to conform with the multivariate normal assumptions of structural equation modelling. In addition, participants with CRP values below the limit of detection (LOD), (i.e., under the value of 1 mg/L) were assigned a value of 1 mg/L.

The path analysis was conducted to model the relationships between the glycaemic markers and each individual inflammatory marker (IL-6, CRP and GlycA) and outcome (WBV, WM volume and GM volume). This model was initially created as a minimally adjusted model (considered age, TIV) (see S1A Fig) but then built into a fully confounder-adjusted model including child and adult SEP, education, BMI, alcohol, smoking status, physical activity, arthritis medication use, and COPD, and total intracranial volume (TIV) and age at scan (S1B Fig).

**Supplementary analysis.** In addition to this, we conducted a latent analysis model where we treated inflammation as an unobserved ("latent") factor that manifests through IL-6, CRP and GlycA. Each biomarker was modelled as the latent inflammation factor multiplied by a factor loading—quantifying how strongly it reflects inflammation—plus a residual term capturing marker-specific noise. By fitting this simple CFA/SEM model, we obtain a single, more reliable inflammation score for each individual, which we then use in downstream analyses instead of juggling three correlated biomarkers. The results were not part of the main. The latent variable model was initially built as a minimally confounder-adjusted model but then constructed with full adjustments for confounders. For the model with a latent variable, a number of fit indices were then explored to test model adequacy, more specifically the Chi-square test, root mean square error of approximation (RMSEA) and the comparative fit index (CFI).

                                                      

We further explored the models

Path analysis sensitivity analyses additionally evaluated alternative systemic inflammation indices, including the neutrophil–lymphocyte ratio (NLR) and the systemic immune-inflammation index (SII), using the same fully adjusted mediation framework (estimating c, c′, a, b, and ab paths) as in the primary models for WBV.

## Results

The sample consisted of 216 women with glycaemic measures at age 60–64 and neuroimaging data. A flowchart of the participants is presented in Fig 1 and sample characteristics are shown in Table 1.

For IL-6 analyses, 10 participants with IL-6 levels above the maximum level reliably quantified by the current methods (10pg/mL) were excluded.

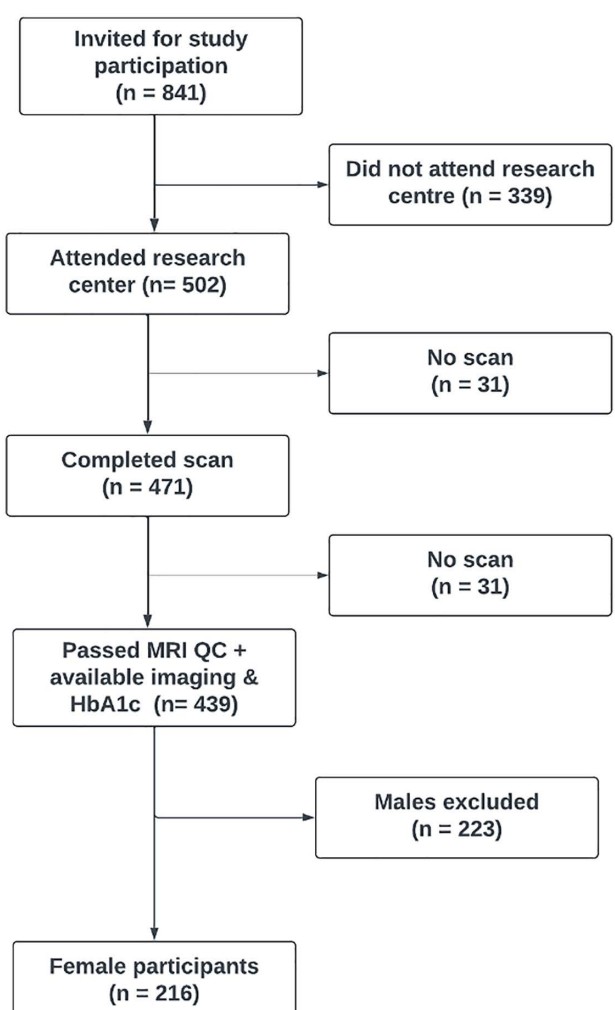

**Fig 1. Flowchart providing an overview of Insight 46 recruitment of National Survey of Health and Development participants who undertook imaging and were part of my study.** To be considered in this study, participants had to have available volumetric imaging data, HbA$_{1c}$ data at age 60-64 and be a female. This amounted to 216 participants being included in the study.

**Table 1. Sample characteristics for the participants considered in the analysis (max n = 216).**

| Participant characteristics | | n | |
|---|---|---|---|
| Standardised childhood cognition score | | 212 | 0·44 (0·74) |
| Education | | 216 | |
| | No qualifications | 30 (14%) | |
| | Below O-levels (vocational) | 18 (8%) | |
| | O-levels and equivalents | 53 (25%) | |
| | A-levels and equivalents | 80 (37%) | |
| | Degree or higher | 35 (16%) | |
| Adult socioeconomic position | | 216 | |
| | Non-manual (Class I–IIIN) | 186 (87%) | |
| | Manual (Class IIIM-V) | 30 (13%) | |
| Childhood socioeconomic position | | 212 | |
| | Non-manual (Class I–IIIN) | 114 (55%) | |
| | Manual (Class IIIM-V) | 98 (45%) | |
| **Metabolic markers at age 60–64** | | | |
| $HbA_{1c}$, %, at age 60–64 | | 216 | 5.8 (0.55) |
| $HbA_{1c}$, mmol/mol | | 216 | 38.8 (6) |
| Glucose mmol/L | | 216 | 5.6 (1.2) |
| Interleukin-6 (IL-6) pg/mL | | 204 | 2.1 (1.4) |
| C-reactive protein (CRP) (mg/L) | | 212 | 3.5 (6.6) |
| GlycA (mg/L) | | 202 | 1.1 (0.3) |
| Diabetes medication use | | 216 | 6 (2.7%) |
| BMI kg/m2 | | 216 | 27.5 (4.9) |
| Smoking status | | 205 | |
| | Current Smokers | 10 (5%) | |
| | Ex-smokers | 70 (34%) | |
| | Never smoker | 125 (61%) | |
| Alcohol (units/week) | | 216 | |
| | ≤ 14 | 198 (91%) | |
| | > 14 | 18 (9%) | |
| Physical activity levels | | 219 | |
| | Inactive | 107 (49%) | |
| | Moderately active | 47 (21%) | |
| | Most Active | 65 (30%) | |
| COPD | | | 124 |
| | Yes | | 9 |
| | No | | 115 |
| Arthritis | | | 213 |
| | Yes | | 2 |
| | No | | 211 |
| **Neuroimaging metrics, age 69–71** | | | |
| Mean age at scanning, years | | 216 | 70.7 (0.7) |
| Whole brain volume (WBV), mL | | 216 | 1046.7 (82.4) |
| White matter volume (WM), mL | | 216 | 394.3 (2.8) |

*(Continued)*

**Table 1.** (Continued)

| Participant characteristics | n | |
|---|---|---|
| Grey matter volume (GM), mL | 216 | 602.6 (3) |
| Total intracranial volume (TIV) | 216 | 1342.4 (91.8) |

Values are n (%), mean (SD) and median (IQR). Whole brain, grey matter and white matter volume measurements reported are unadjusted for total intercranial volumes for these descriptions. % are calculated against the max data available for that specific measure for the pooled sample. SD: Standard deviation. As described above, to be considered in the study, participants had to have available volumetric imaging data, HbA1c data at age 60–64 and be a female which amounted to 216 participants.

## Associations between inflammatory markers

The correlation matrix shows weak associations between the different inflammatory markers. The associations were strongest between CRP and IL-6 and least strong between GlycA and CRP. There were positive relationships between IL6, glycA and CRP. The relationship between IL6 and CRP was the strongest (r = 0.4, p < 0.001), followed by the association between glycA and CRP (r = 0.2, p = 0.004) and GlycA and IL6 (r = 0.1, p = 0.01).

**Path analysis of HbA$_{1c}$ and brain structure.** As discussed in the methods, the results presented here are the path analysis for the fully confounder-adjusted models.

*Whole brain volumes*: There was a total effect of higher HbA1c at age 60–64 on lower WBV at age 69–71 for all the models (see Fig 2).

In the mediation model, the link between HbA1c and whole-brain volume remained essentially a direct one, even after accounting for CRP, IL-6 and GlycA. There was no indication that any of those inflammatory markers carried part of the effect between HbA1c and brain volume. Separately, higher HbA1c was associated with elevations in IL-6 and GlycA, but it showed no meaningful relationship with CRP.

*Grey matter volumes*: Across all three inflammatory-marker models (CRP, IL-6, GlycA), higher HbA$_1$c was associated with reduced gray-matter volume, with total effect estimates of approximately (see Fig 3). When CRP, IL-6 or GlycA were added to the model, the HbA$_1$c→GM relationship remained essentially unchanged, and none of the markers mediated this relationship (indirect effects very close to zero, all p > 0.6) (see Fig 3). Finally, HbA$_1$c was itself positively related to IL-6 and GlycA (p ≤ 0.005) but showed no meaningful link with CRP (p ≈ 0.9).

*White matter volumes*: Since there was no total effect of HbA$_{1c}$ on WM volumes, the results for these analyses were not decomposed into direct and indirect effects. For completeness, the tables are shown in (Fig 4).

**Path analysis of glucose and brain structure.** The findings for glucose on WBV, GM and WM volumes were similar to those for HbA$_{1c}$ (Table 2). For WBV, GM and WM volumes, there was a total effect of glucose on WBV for CRP, IL-6, and GlycA models, with direct effects observed across these markers. However, there was no evidence of indirect effects through the inflammatory pathways. The exposure-mediator pathway was not statistically significant for CRP but was for IL-6 and GlycA. There was no evidence for the presence of a mediator-outcome pathway for any of the inflammatory markers. Overall, the trends observed for glucose were consistent with those seen for HbA$_{1c}$, suggesting the role of both glycaemic measures do not involve indirect inflammatory pathways (Table 2).

## Supplementary analyses

For the latent model of inflammatory markers, there was evidence of a total and direct effect of with HbA$_{1c}$ and glucose on WBV, but no total or direct effect on GM or WM volumes (S1 Table). There was no evidence of indirect effects observed between HbA$_{1c}$ or glucose on WBV, GM, or WM volumes through the inflammatory pathway.

In sensitivity analyses using NLR and SII, the total association with whole brain volume remained negative and statistically significant, but indirect (mediated) effects were small and non-significant (S2 Table), providing no evidence of mediation via these indices.

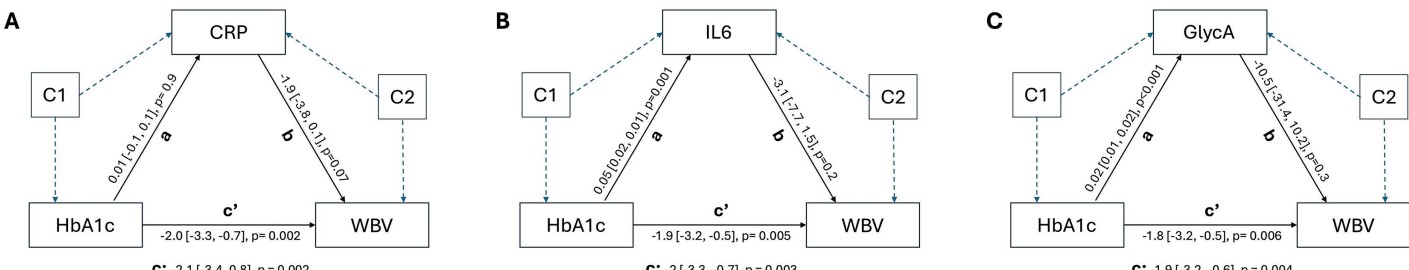

**Fig 2. Mediation model path of HbA1c (exposure), inflammatory marker (mediator) and outcome (whole brain volumes).** Model A considers CRP as the mediator. Model B considers IL-6 as the mediator. Model C considers GlycA as the mediator. As seen in each model: path **a** is the exposure→mediator link adjusted for exposure–mediator confounders. path **b** is the mediator→outcome link adjusted for mediator-outcome confounders. path **c** is the total exposure→outcome effect before including the mediator. path **c′** is the direct exposure→outcome effect after including the mediator. Point estimates, 95% confidence intervals, and *p*-values are shown for every path, with individual covariates collapsed into the two blocks **C1** (exposure-mediator confounders) and **C2** (mediator outcome associations) to aid visual clarity. Exposure-mediator confounders: Age, socioeconomic status, smoking, alcohol, body mass index and education. Mediator-outcome confounders: Age, socioeconomic status, smoking, alcohol, body mass index and education chronic obstructive pulmonary dysfunction and arthritis.

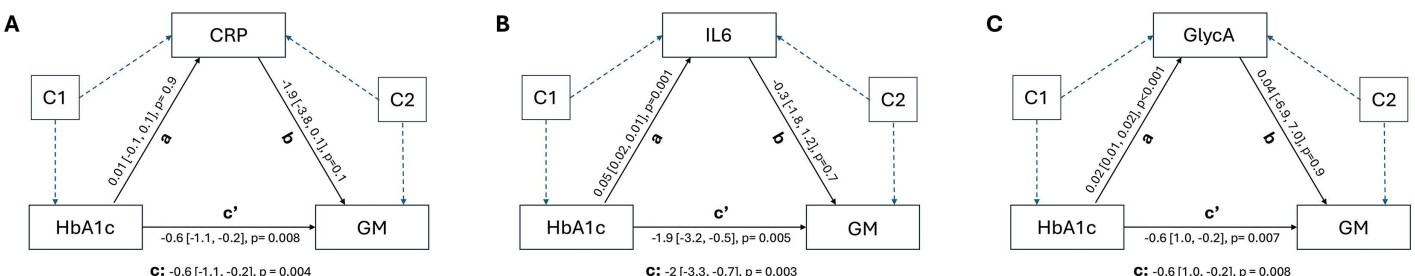

**Fig 3. Mediation model path of HbA1c (exposure), inflammatory marker (mediator) and outcome (gray matter volumes).** Model A considers CRP as the mediator. Model B considers IL-6 as the mediator. Model C considers GlycA as the mediator. As seen in each model: path **a** is the exposure→mediator link adjusted for exposure–mediator confounders. path **b** is the mediator→outcome link adjusted for mediator-outcome confounders. path **c** is the total exposure→outcome effect before including the mediator. path **c′** is the direct exposure→outcome effect after including the mediator. Point estimates, 95% confidence intervals, and *p*-values are shown for every path, with individual covariates collapsed into the two blocks **C1** (exposure-mediator confounders) and **C2** (mediator outcome associations) to aid visual clarity. Exposure-mediator confounders: Age, socioeconomic status, smoking, alcohol, body mass index and education. Mediator-outcome confounders: Age, socioeconomic status, smoking, alcohol, body mass index and education chronic obstructive pulmonary dysfunction and arthritis.

## Discussion

### Summary of findings

The aim of this mediation analysis was to gain better insight into an important potential mechanism linking hyperglycaemia with smaller brain volumes (Fatih et al, 2022). The path analyses demonstrated that there was: 1) a total effect of glycaemic markers on smaller brain structure, 2) hyperglycaemia was mostly associated with higher systemic inflammation (for IL-6 and GlycA) but 3) there we found no association between the systemic inflammatory markers and volumetric brain measures. Hence overall there was no indirect effect of glycaemia on these brain outcomes via a systemic inflammation pathway. Considering the inflammatory markers as a latent variable for inflammation had little influence on the results nor did considering fasting glucose measure as the exposure of any analyses. Overall, the findings suggested that while both high glucose and increased inflammation are related, systemic inflammation, as measured by serum biomarkers, does not mediate the relationship between glucose levels and smaller brain volume in this sample of women.

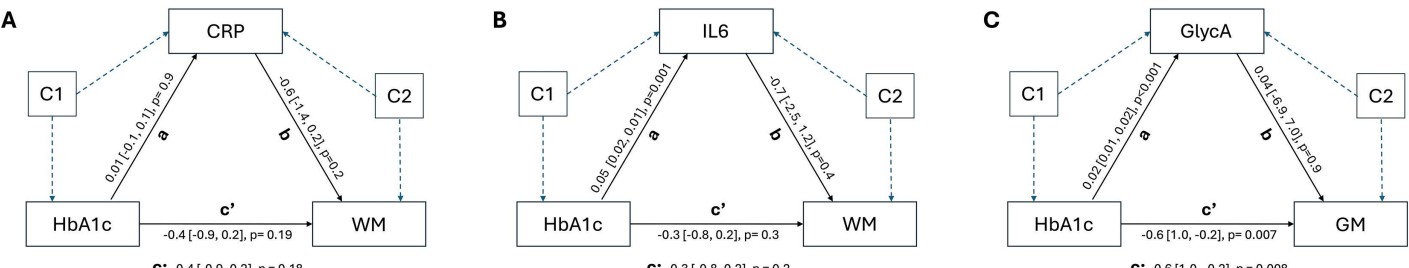

**Fig 4. Mediation model path of HbA1c (exposure), inflammatory marker (mediator) and outcome (white matter volumes).** Model A considers CRP as the mediator. Model B considers IL-6 as the mediator. Model C considers GlycA as the mediator. As seen in each model: path **a** is the exposure→mediator link adjusted for exposure–mediator confounders. path **b** is the mediator→outcome link adjusted for mediator-outcome confounders. path **c** is the total exposure→outcome effect before including the mediator. path **c′** is the direct exposure→outcome effect after including the mediator. Point estimates, 95% confidence intervals, and *p*-values are shown for every path, with individual covariates collapsed into the two blocks **C1** (exposure-mediator confounders) and **C2** (mediator outcome associations) to aid visual clarity. Exposure-mediator confounders: Age, socioeconomic status, smoking, alcohol, body mass index and education. Mediator-outcome confounders: Age, socioeconomic status, smoking, alcohol, body mass index and education chronic obstructive pulmonary dysfunction and arthritis.

**Table 2. Path analysis of the fully confounder-adjusted models of the fasting glucose-brain associations (whole brain, grey matter and white matter volumes) via inflammation (c-reactive protein, interleukin-6 and glycoprotein-A).**

**Whole brain volumes (WBV)**

|  | | c-reactive protein (CRP) | | | interleukin-6 (IL-6) | | | glycoprotein-A (GlycA) | | |
|---|---|---|---|---|---|---|---|---|---|---|
|  | Path | β | 95% CI | p | β | 95% CI | p | β | 95% CI | p |
| Total effect | c | −6.5 | −11.4  −1.6 | 0.009 | −6.5 | −11.4  −1.6 | 0.009 | −6.4 | −11.3  −1.5 | 0.011 |
| Direct effect | c′ | −6.4 | −11.3  −1.5 | 0.011 | −6.1 | −11.0  −1.2 | 0.013 | −6.0 | −11.0  −1.0 | 0.021 |
| Indirect effect | ab | −0.1 | −0.8  0.5 | 0.723 | −0.4 | −1.1  0.3 | 0.366 | −0.4 | −1.5  0.6 | 0.386 |
| Exposure-mediator | a | 0.1 | −0.3  0.5 | 0.711 | 0.1 | −0.04  0.3 | 0.102 | 0.05 | 0.01  0.1 | 0.005 |
| Mediator-outcome | b | −2.0 | −3.5  0.4 | 0.138 | −3.4 | −8.1  1.3 | 0.092 | −9.4 | −30.6  11.6 | 0.381 |

**Grey matter volumes (GM)**

|  | | c-reactive protein (CRP) | | | interleukin-6 (IL-6) | | | glycoprotein-A (GlycA) | | |
|---|---|---|---|---|---|---|---|---|---|---|
|  | Path | β | 95% CI | p | β | 95% CI | p | β | 95% CI | p |
| Total effect | c | −1.7 | −3.4  −0.09 | 0.044 | −1.7 | −3.4  −0.05 | 0.041 | −1.7 | −3.4  −0.04 | 0.042 |
| Direct effect | c′ | −1.7 | −3.4  −0.06 | 0.042 | −1.6 | −3.3  −0.04 | 0.049 | −1.7 | −3.4  −0.02 | 0.041 |
| Indirect effect | ab | −0.03 | −0.2  0.1 | 0.724 | −0.05 | −0.2  0.1 | 0.631 | −0.001 | −0.3  0.3 | 0.902 |
| Exposure-mediator | a | 0.1 | −0.3  0.5 | 0.738 | 0.1 | −0.04  0.3 | 0.112 | 0.05 | 0.01  0.08 | 0.005 |
| Mediator-outcome | b | −0.4 | −1.0  0.2 | 0.233 | −0.4 | −1.9  1.0 | 0.512 | −0.03 | −7.1  7.0 | 0.912 |

**White matter volumes (WM)**

|  | | c-reactive protein (CRP) | | | interleukin-6 (IL-6) | | | glycoprotein-A (GlycA) | | |
|---|---|---|---|---|---|---|---|---|---|---|
|  | Path | β | 95% CI | p | β | 95% CI | p | β | 95% CI | p |
| Total effect | c | −2.3 | −4.3  −0.3 | 0.029 | −2.2 | −4.2  −0.2 | 0.032 | −2.2 | −4.2  −0.2 | 0.031 |
| Direct effect | c′ | −2.2 | −4.2  −0.3 | 0.032 | −2.2 | −4.2  −0.1 | 0.042 | −2.0 | −4.0  −0.3 | 0.050 |
| Indirect effect | ab | −0.05 | −0.3  0.2 | 0.721 | −0.01 | −0.3  0.2 | 0.592 | −0.2 | −0.7  0.2 | 0.332 |
| Exposure-mediator | a | 0.1 | −0.3  0.5 | 0.702 | 0.1 | −0.04  0.3 | 0.123 | 0.05 | 0.01  0.08 | 0.006 |
| Mediator-outcome | b | −0.6 | −1.4  0.2 | 0.142 | −0.7 | −2.5  1.1 | 0.442 | −5.3 | −13.7  3.1 | 0.211 |

The table presents the β coefficients, confidence intervals and p values. The models were adjusted for TIV, age at scan, socioeconomic position, education, alcohol, smoking, physical activity, BMI, arthritis and COPD.

## Specific findings and concordance with the literature

The study was motivated by previous research suggesting that individuals with T2D show elevated levels of inflammatory markers compared to healthy controls (regardless of disease duration) [28]. Similarly, women with T2D showed higher inflammation TNF-α, IL-6 and CRP compared to women without the condition [29,30]. There is also evidence that inflammatory markers are associated with poorer brain health and that cytokines accumulate at different rates in AD patients compared with healthy control subjects [6,7,31].

Our observation that poorer glycaemia was associated with higher inflammation, is in line with previous evidence in a mixed sample of individuals with T2D and healthy controls [32]. This is consistent with the known effect of hyperglycaemia on NF-κB–dependent inflammatory cytokine production and other mechanisms linked to the activation of inflammatory pathways [4,33].

The evidence linking inflammation to brain health is less consistent. Previous analyses from population-based studies have revealed mixed findings in relation to the association between markers of systemic inflammation and brain health outcomes. For example, recent findings from the Atherosclerosis Risk in Communities (ARIC) study failed to find an association between inflammatory markers and WBV [7] consistent with our findings. In contrast, the Framingham study found that most, but not all, of the inflammatory markers they considered were associated with lower total brain volume [34].

Discrepancy in the results to the Framingham study may be explained by the different methodological approaches taken. Firstly, Framingham considered a mixed-sex sample and, notably, reported stronger inverse inflammation-brain association in men, with the corresponding slopes in women were smaller and often non-significant. For example, the association between IL-6 and their brain outcome ranged in the null in women (akin to ours) but was robust in men.

In our analysis, we only considered women and had a narrower biomarker scope (only available data for CRP, IL-6 and GlycA) comparatively to Framingham who assayed ten cytokines, allowing robust signals from markers such as OPG and TNF-α to emerge. Despite this, our findings for IL-6 were largely consistent with those of their study. Framingham also considered a head-size-normalised total cerebral brain volume (TCBV) measured in very close temporal proximity to their biomarker (biomarker and scan on the same visit). We linked our inflammatory biomarkers measured at ages 60–64 to raw whole-, grey- and white-matter volumes ten years later, then fitted mediation model. Taken together—our women-only focus, the narrow biomarker panel, the wider (ten-year) interval between biomarker assessment and MRI, the use of absolute brain-volume measures, and different confounder adjustment may explain differences in the findings. Since their findings were less notable in their women cohort (and at times consistent with ours, e.g., IL-6), it may be argued that the absence of a significant effect in NSHD is compatible with, rather than contradictory to, the modest negative trend reported for women in the Framingham cohort.

There remains an important question, that is whether markers of systemic inflammation can give us a robust insight into brain inflammation. Some previous studies have found that systemic inflammation, particularly through infections, can influence neuroinflammatory processes, but the interplay is complex [35]. Systemic IL-6 and other cytokines contribute to immune changes in the brain, yet these effects are modulated by local brain mechanisms and do not necessarily follow a linear systemic-to-brain pathway. For instance, systemic infections can elevate arterial cytokine levels (e.g., IL-6, IL-8, IL-10) while decreasing certain cytokines in the brain's extracellular fluid [35]. This intricate relationship highlights the challenges of relying solely on systemic inflammatory markers to understand what may be happening in the brain and underscores the need for more precise biomarkers to elucidate the impact of neuroinflammation on brain health and neurological disorders.

Interestingly, we observed an association between glycaemia and GlycA and IL-6 not CRP. Mechanistically, studies have shown that IL-6 and CRP are closely related, with IL-6 being the major factor that triggers the hepatic synthesis of CRP [36,37]. However, although inflammatory markers are usually linked, recent studies have found them to diverge in certain contexts. For example, differing concentrations of IL-6 and CRP have been found in relation to HRT use [38]. Similarly, divergent levels of these inflammatory markers have been found in relation to other clinical factors such as alcohol

use and exercise [38]. Thus, although CRP and IL-6 are biologically linked, their levels can diverge under certain conditions which may account for the weak correlation observed in this analysis.

Considering inflammation as a latent variable offered no additional insight into these associations (see S1 Table). When designing this study, we made the decision to construct a latent variable for inflammation using IL-6, CRP and GlycA. The rationale was that combining multiple measures may give a more comprehensive insight into systemic inflammation by capturing the common variance amongst them and reducing measurement error. However, the general lack of correlation between the inflammatory variables may have limited the potential utility in line with the fit indices suggesting some inconsistency in terms of the model's adequacy.

It is possible that the relationship between hyperglycaemia and brain health is mediated by factors other than inflammation. While we found no evidence that systemic inflammation directly mediates the relationship between HbA$_{1c}$ and WBV, other possible mediators to consider may be oxidative stress, small vessel disease and CVD [4]. Other factors that may be particularly important around late midlife include hormonal health such as oestrogen and its neuroprotective effect reducing during the perimenopause and the post menopause. This change in hormonal health can influence insulin sensitivity and glucose metabolism [3] as well as brain structure and function in women [39] Additionally, non-biological factors in social roles and responsibilities, including caregiving duties and family responsibilities, may mediate these sex-specific associations found in women by imposing chronic stress and time constraints, which might affect brain health. Future research should consider important biological and social factors specific to women to achieve a comprehensive understanding of the underlying mechanisms of the glycaemia-brain health pathways.

In this study, we considered that hyperglycaemia precedes inflammation. This is based on previous studies suggesting that hyperglycaemia and abnormal glucose metabolism can result in the formation of advanced glycation end products (AGES) with reactive oxygen species (ROS)triggers pathways that regulate the inflammatory response resulting in the increase of pro-inflammatory cytokines such as IL-6 [4]. However, these cytokines produced by adipose tissue and macrophages may also result in a state of IR thus contributing to the pathophysiology of T2D [40] This was found even when adjustments were made for inflammatory-related confounders (e.g., smoking, exercise, and BMI). In an attempt to reduce the likelihood of infection-driven hyperglycaemia, participants with an IL-6 value of > 10 pg/mL were excluded prior to my analyses.

## Strengths and weaknesses

The study has multiple strengths. First, it considered multiple markers of inflammation. Few population-based studies have had the availability of multiple inflammatory markers. The confounder data includes those for the exposure-mediator and the mediator-outcome paths, which are known problems with causal mediation studies [27]. Furthermore, an important strength of this study is that it considered samples of women of the same age. The homogeneity of age ensured age-specific brain changes did not confound the results, providing a clearer picture of the relationship between HbA$_{1c}$ and brain health.

Since inflammatory markers were not associated with the brain imaging measures considered, further studies should consider their associations to more subtle markers of brain pathology such as those of microstructural diffusion metrics and enlarged periventricular spaces. This will provide an insight into the impact of diabetes-related inflammation on early-stage small vessel disease.

It is worth acknowledging the potential for reverse causation, where changes in brain volume could influence metabolic parameters, rather than the reverse. This highlights the need for caution in interpreting the directionality of the observed relationships. Additionally, other unmeasured variables not considered in this analysis, such as genetic factors or specific dietary components, might influence the relationships explored. Several other limitations should be considered. First, inflammatory biomarkers were quantified using ELISA. Compared with higher-sensitivity multiplex or proteomic platforms (e.g., Luminex or Olink), ELISAs may have lower sensitivity and dynamic range for low-abundance analytes and can be more susceptible

to inter-assay variability. Consequently, non-differential measurement error may have attenuated observed associations. Second, biomarkers were measured at a single assessment, which may not capture longer-term inflammatory burden or within-person variability over time; this could further dilute associations and limits inference about trajectories. Third, as an observational study, residual confounding cannot be excluded despite adjustment for measured covariates (e.g., unmeasured health behaviours, subclinical disease, medication use, or acute infections at sampling). Relatedly, reverse causation remains possible if preclinical processes influenced inflammatory profiles prior to outcome ascertainment. Fourth, selection and attrition may affect generalisability: participants who attended clinic assessments and provided blood samples may differ systematically from those who did not. Fifth, the MRC National Survey of Health and Development (NSHD) is a single-year birth cohort (all participants born in 1946), and findings may therefore reflect cohort- and period-specific exposures across the life course (e.g., early-life environment, healthcare, smoking patterns, and socioeconomic conditions) rather than age effects alone; replication in other birth cohorts and more contemporary populations will be important.

## Conclusions

These findings reveal that for women of this birth cohort, the relationship between glycaemia in midlife and later life smaller brains was not mediated by systemic inflammation as measured by selected blood markers. As per the findings we reported in our previous paper, poorer glycaemia is directly associated with smaller brains but there was no indirect path of this relationship through inflammation. Future studies could investigate these associations in a sample such as UK Biobank to examine the causal the nature of these relationships through techniques such as Mendellian Randomisation.

## Supporting information

**S1 File. Supplementary information.** Contains supplementary figures and tables detailing mediation models and inflammation-based pathway analyses relating glycaemic markers to brain volume outcomes.
(DOCX)

**S1 Fig. Path analysis models conducted in the analyses A) Simple mediation model as a first step of model building.** B) Fully confounder-adjusted model. Adjustments were made for both exposure-mediator and mediator-outcome relationships.
(PNG)

**S2 Fig. A theoretical analysis of the mediation models considered.** A) shows the assumed (total) effect, c. B) the total effect decomposed into a direct, c' and indirect (mediated via a and b) effect.
(PNG)

**S1 Table. Path analysis output of the latent model for the HbA$_{1c}$ and glucose- brain associations (whole brain, grey matter and white matter volumes) via the latent variable of inflammation (constructed using c-reactive protein, interleukin-6 and glycoprotein-a).** The total effects, direct, indirect effects, exposure-mediator are presented. β, CI and p-values are presented.
(DOCX)

**S2 Table. Mediation analyses for whole brain volume (WBV) with inflammation indices neutrophil-to-lymphocyte ratio (NLR) and systemic immune-inflammation index (SII; platelets × neutrophils/ lymphocytes) as candidate mediators.** Shown are the total effect (c), direct effect (c'), indirect effect (ab), and component paths a (exposure→mediator) and b (mediator→WBV), with β coefficients, 95% confidence intervals (CI), and p-values (p). For both NLR and SII, total and direct effects were negative and statistically significant, whereas indirect effects were not significant, providing little evidence that NLR or SII mediates the exposure–WBV association.
(DOCX)

                                                                

## Acknowledgments

We are very grateful to those study members who helped in the design of the study through focus groups, to past and present members of the National Survey of Health and Development study team who helped to collect the data and to the participants both for their contributions to Insight 46 and for their commitments to research over the last seven decades.

## Author contributions

**Conceptualization:** Nasri fatih.

**Formal analysis:** Nasri fatih.

**Funding acquisition:** Marcus Richards, Jonathan M Schott.

**Investigation:** Nasri fatih.

**Methodology:** Nasri fatih, Carole H. Sudre, Richard J Silverwood, David M Cash, Ian B Malone, Josephine Barnes.

**Project administration:** Nasri fatih.

**Resources:** Nasri fatih, Nish Chaturvedi.

**Software:** Nasri fatih, Carole H. Sudre.

**Supervision:** Nish Chaturvedi, Victoria Garfield, Richard J Silverwood, Alun D Hughes, Sarah-Naomi James.

**Visualization:** Nasri fatih.

**Writing – original draft:** Nasri fatih.

**Writing – review & editing:** Nasri fatih, Nish Chaturvedi, Victoria Garfield, Carole H. Sudre, Richard J Silverwood, David M Cash, Josephine Barnes, Marcus Richards, Jonathan M Schott, Alun D Hughes, Sarah-Naomi James.

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
