## [Decision Letter · Decision Letter 0]

12 Jan 2026

Dear Dr. fatih,

Thank you for submitting your manuscript to PLOS ONE. After careful consideration, we feel that it has merit but does not fully meet PLOS ONE’s publication criteria as it currently stands. Therefore, we invite you to submit a revised version of the manuscript that addresses all of the reviewer's points raised during the review process.

We look forward to receiving your revised manuscript.

Kind regards,

Pratibha V. Nerurkar, Ph.D

Academic Editor

PLOS One

Journal Requirements:

3. Please note that your Data Availability Statement is currently missing the DOI/accession number of each dataset OR a direct link to access each database. If your manuscript is accepted for publication, you will be asked to provide these details on a very short timeline. We therefore suggest that you provide this information now, though we will not hold up the peer review process if you are unable.

4. We notice that your supplementary figures are uploaded with the file type 'Figure'. Please amend the file type to 'Supporting Information'. Please ensure that each Supporting Information file has a legend listed in the manuscript after the references list.

Reviewers' comments:

Reviewer's Responses to Questions

**Comments to the Author**

1. Is the manuscript technically sound, and do the data support the conclusions?

Reviewer #1: Yes

Reviewer #2: Yes

2. Has the statistical analysis been performed appropriately and rigorously?

Reviewer #1: Yes

Reviewer #2: Yes

3. Have the authors made all data underlying the findings in their manuscript fully available?

Reviewer #1: No

Reviewer #2: Yes

4. Is the manuscript presented in an intelligible fashion and written in standard English?

Reviewer #1: Yes

Reviewer #2: Yes

Reviewer #1: In this article, the authors investigate the mediating effect of inflammatory biomarkers on the relationship between glycaemic markers and brain health outcomes in women. They did not find evidence for a significant mediating effect of these inflammatory biomarkers.

Overall, this was a well-written paper with sufficient methodological and statistical detail. I have a few suggestions for improvement:

1) Line 69 has an incomplete sentence

2) Line 108 was slightly confusing, since if the participants were all born in the same week and the postal questionnaires were all sent in 2006, they should all have roughly the same age, not a range of 60-64. I assume this is because postal questionnaires were actually sent out over the course of 4 years? If so, this should be made clear.

3) It would be helpful if p values are reported with standard formatting, i.e. up to the third decimal number.

4) Page 25 – the discussion of study limitations is very sparse. I would recommend including at least a few other notable limitations of the study, for instance the use of ELISAs for measuring inflammatory biomarkers (as opposed to Luminex or Olink, which have greater sensitivity).

Reviewer #2: Overall, this is a methodologically rigorous manuscript addressing an interesting and important topic. My only question is how indicators such as NLR and SII would impact the results. Also what test was done to assure the readers that missing values were random in nature verse MNAR. Presumably, few values were missing?

**Do you want your identity to be public for this peer review?** For information about this choice, including consent withdrawal, please see our Privacy Policy

Reviewer #1: No

Reviewer #2: No

---

## [Author Response · Author response to Decision Letter 1]

20 Feb 2026

Dear Pratibha,

We are grateful of your feedback on our manuscript " Examining the role of systemic inflammation as a mediator of the glycaemia-brain volume associations in women".

We have implemented the feedback you and reviewers have shared and would like to submit a revised version of the manuscript with changes. For your convenience, we have listed each of the editor and reviewer comments below and included details on how we have addressed them all in the revised manuscript.

Addressing editor comments:

Comment: Thank you for providing this Data Availability Statement. Could you please provide a website for the NSHD data-sharing platform (Skylark) where a researcher could request access to this data?

Response: We have now added a link to the data-sharing platform Skylark through the manuscript and submission steps. The paragraph and link read as follow: "Data from the MRC National Survey of Health and Development (NSHD) are curated by the MRC Unit for Lifelong Health and Ageing at UCL and are available to bona fide researchers via application through the NSHD data-sharing platform (Skylark: www.skylark.ucl.ac.uk/NSHD). An NSHD dataset is also deposited with the UK Data Service (DOI: 10.5255/UKDA-SN-8732-3)".

Comment: Please ensure that your manuscript meets PLOS ONE's style requirements, including those for file naming. The PLOS ONE style templates can be found at

Response: We have followed the formatting guidelines you sent us to ensure that our manuscript aligns with PLOS ONE’s style requirements.

Comment: We note that the grant information you provided in the ‘Funding Information’ and ‘Financial Disclosure’ sections do not match.

Response: We have reviewed and ensured that the funding information and financial disclosure match. The financial disclosure reads as follow now:

Financial disclosure: This work was supported by Alzheimer’s Research UK (awards ARUK-PG2014-1946 and ARUK-PG2017-1946 to Jonathan Schott; additional support to Josephine Barnes), Alzheimer Society (award AS-JF-17-011 to Carole Sudre), UCL Queen Square Institute of Neurology, University College London (support to Jonathan Schott), Diabetes Research and Wellness Foundation (award SCA/01/NCF/22 to Victoria Garfield) , Medical Research Council funding for the Medical Research Council National Survey of Health and Development (MC_UU_12019/1), British Heart Foundation (PG/17/90/33415) to Alun Hughes. Alun Hughes also receives support from the Horizon Europe Programmes of the European Union through Innovate UK (HORIZON-RIA 10113672), the Wellcome Trust (221774/Z/20/Z), a British Heart Foundation Centre of Research Excellence award to UCL (RE/24/130013) and the National Institute for Health Research University College London Hospitals Biomedical Research Centre.

Comment: Please note that your Data Availability Statement is currently missing the DOI/accession number of each dataset OR a direct link to access each database.

Response: We have now added a direct link the NSHD/Insight 46 dataset in the Data Availability Statement. It now reads as follow: “Data from the MRC National Survey of Health and Development (NSHD) are curated by the MRC Unit for Lifelong Health and Ageing at UCL and are available to bona fide researchers via application through the NSHD data-sharing platform (Skylark). An NSHD dataset is also deposited with the UK Data Service (DOI: 10.5255/UKDA-SN-8732-3).”

Reviewer: We notice that your supplementary figures are uploaded with the file type 'Figure'. Please amend the file type to 'Supporting Information'

Response: We have now reuploaded the supplementary figures under the file type “Supporting Information”.

Response: Please review your reference list to ensure that it is complete and correct. If you have cited papers that have been retracted, please include the rationale for doing so in the manuscript text, or remove these references and replace them with relevant current references. Any changes to the reference list should be mentioned in the rebuttal letter that accompanies your revised manuscript. If you need to cite a retracted article, indicate the article’s retracted status in the References list and also include a citation and full reference for the retraction notice.

Response: We have checked our reference list to ensure its complete and correct.

Addressing: reviewer 1 comments:

Reviewer 1 comment: Line 69 has an incomplete sentence

Response: We have completed this sentence on page 3. It now reads as follow: This aligns with other studies which also reported sex-specific differences in diabetes-related mechanisms and health outcomes [2]

Reviewer 1 comment: Line 108 was slightly confusing, since if the participants were all born in the same week and the postal questionnaires were all sent in 2006, they should all have roughly the same age, not a range of 60-64. I assume this is because postal questionnaires were actually sent out over the course of 4 years?

Response: Thank you for the comment. Although participants were of very similar age, clinic appointments were conducted over an extended period due to scheduling and operational constraints, so visits occurred across ages 60–64. We have updated the wording to clarify that the clinic visit window spanned multiple years. It now reads as follows: “In 2006, study members were sent postal questionnaires. They were also invited to attend a clinic visit between ages 60 and 64 (conducted over multiple years), during which blood samples were collected to assess glycaemic health and systemic inflammation”.

Reviewer 1 comment: It would be helpful if p values are reported with standard formatting, i.e. up to the third decimal number.

Response: Thank you for pointing this out. We now report the p values up to the third decimal points throughout the manuscript.

Reviewer 1 comment: Page 25 – the discussion of study limitations is very sparse. I would recommend including at least a few other notable limitations of the study, for instance the use of ELISAs for measuring inflammatory biomarkers (as opposed to Luminex or Olink, which have greater sensitivity).

Response: Thank you for the suggestion. We have expanded our limitation section to include additional methodological limitations. In particular, we now note that inflammatory biomarkers were quantified using ELISA; compared with higher-sensitivity multiplex or proteomic platforms (e.g., Luminex or Olink), ELISAs may have lower sensitivity and dynamic range for low-abundance analytes and can be more susceptible to inter-assay variability, which may introduce measurement error and attenuate associations. We also added limitations relating to (i) reliance on single time-point biomarker measurements, (ii) potential residual confounding and reverse causation inherent to the observational design, and (iii) generalisability to populations beyond the study sample.

It now reads as follow: “Several other limitations should be considered. First, inflammatory biomarkers were quantified using ELISA. Compared with higher-sensitivity multiplex or proteomic platforms (e.g., Luminex or Olink), ELISAs may have lower sensitivity and dynamic range for low-abundance analytes and can be more susceptible to inter-assay variability. Consequently, non-differential measurement error may have attenuated observed associations. Second, biomarkers were measured at a single assessment, which may not capture longer-term inflammatory burden or within-person variability over time; this could further dilute associations and limits inference about trajectories. Third, as an observational study, residual confounding cannot be excluded despite adjustment for measured covariates (e.g., unmeasured health behaviours, subclinical disease, medication use, or acute infections at sampling). Relatedly, reverse causation remains possible if preclinical processes influenced inflammatory profiles prior to outcome ascertainment. Fourth, selection and attrition may affect generalisability: participants who attended clinic assessments and provided blood samples may differ systematically from those who did not. Fifth, the MRC National Survey of Health and Development (NSHD) is a single-year birth cohort (all participants born in 1946), and findings may therefore reflect cohort- and period-specific exposures across the life course (e.g., early-life environment, healthcare, smoking patterns, and socioeconomic conditions) rather than age effects alone; replication in other birth cohorts and more contemporary populations will be important.”

Addressing: reviewer 1 comments:

Reviewer 2 comment: Also what test was done to assure the readers that missing values were random in nature verse MNAR. Presumably, few values were missing?

Response: Thank you for raising this point. We agree the missing-data mechanism is important. Missingness was low overall in the analytic sample (n = 216), with complete data for volumetric imaging and HbA1c (0% missing), and a maximum of 6.5% missingness across other model variables (GlycA: 202/216; IL-6: 204/216; smoking: 205/216; other covariates typically ≥212/216). We used full information maximum likelihood (FIML), which provides valid inference under a Missing At Random (MAR) assumption. As noted in Rubin’s missing-data framework, MAR versus Missing Not At Random (MNAR) is generally not testable using observed data alone without additional assumptions or external information (Rubin, 1976; Little & Rubin, 2002). Accordingly, we assessed the plausibility of MAR by comparing observed baseline characteristics between participants with complete versus incomplete data. This check did not indicate strong systematic missingness with respect to key observed variables.

Reviewer 2 comment: My only question is how indicators such as NLR (Neutrophil-to-Lymphocyte Ratio) and SII (Systemic Immune-Inflammation Index) would impact the results

Response: Thank you for the suggestion to evaluate additional inflammation indices derived from routine blood counts, specifically the neutrophil-to-lymphocyte ratio (NLR) and the systemic immune-inflammation index (SII). In response, we ran the same mediation framework used for CRP and IL-6 with NLR and SII as alternative mediators to whole brain volumes and have added these results to the Supplementary Materials (Supplementary Table 2).

Overall, the findings using NLR and SII were consistent with our primary analyses. For whole brain volume, the total association remained negative and statistically significant (NLR: c = −6.64, p = 0.012; SII: c = −6.74, p = 0.023), and the direct effect estimates were very similar in magnitude (NLR: c′ = −6.42, p = 0.011; SII: c′ = −6.63, p = 0.022). There was no evidence of mediation via either index: indirect effects were small and non-significant (NLR: ab = −0.24, p = 0.831; SII: ab = −0.11, p = 0.901), and neither the exposure→mediator (a) nor mediator→outcome (b) paths were statistically significant (all p ≥ 0.13). We therefore conclude that incorporating NLR and SII does not materially change the interpretation: results are consistent across inflammatory markers/indices, with little evidence that the association is mediated through these measures. I have added a brief line in the methods section of the manuscript, under supplementary analyses, stating that these analyses were conducted and a brief line in the results summarising these findings. The table can be found in Supplementary Table 2

We hope the revised manuscript is now suitable for publication and would be pleased to address any additional comments.

Kind regards,

Nasri Fatih, PhD (on behalf of all co-authors)

---

## [Editor Report · Decision Letter 1]

25 Feb 2026

Examining the role of systemic inflammation as a mediator of the glycaemia-brain volume associations in women

PONE-D-25-37103R1

Dear Dr. fatih,

We’re pleased to inform you that your manuscript has been judged scientifically suitable for publication and will be formally accepted for publication once it meets all outstanding technical requirements.

Kind regards,

Pratibha V. Nerurkar, Ph.D

Academic Editor

PLOS One
---

## [Editor Report · Acceptance letter]

PONE-D-25-37103R1

PLOS One

Dear Dr. fatih,

I'm pleased to inform you that your manuscript has been deemed suitable for publication in PLOS One. Congratulations! Your manuscript is now being handed over to our production team.

Kind regards,

on behalf of

Dr. Pratibha V. Nerurkar

Academic Editor

PLOS One